# The Landscape of Stereotactic Ablative Radiotherapy (SABR) for Renal Cell Cancer (RCC)

**DOI:** 10.3390/cancers16152678

**Published:** 2024-07-27

**Authors:** Elena Moreno-Olmedo, Ami Sabharwal, Prantik Das, Nicola Dallas, Daniel Ford, Carla Perna, Philip Camilleri

**Affiliations:** 1GenesisCare, Oxford OX4 6LB, UK; ami.sabharwal@genesiscare.co.uk (A.S.); prantik.das@genesiscare.co.uk (P.D.); nicola.dallas@genesiscare.co.uk (N.D.); daniel.ford@genesiscare.co.uk (D.F.); carala.perna@genesiscare.co.uk (C.P.); philip.camilleri@genesiscare.co.uk (P.C.); 2Department of Radiotherapy and Oncology, Oxford University Hospitals NHS Foundation Trust, Oxford OX3 7LE, UK; 3Department of Oncology, Royal Derby Hospital, Derby DE22 3NE, UK; 4Department of Oncology, Royal Berkshire Hospital, Reading RG1 5AN, UK; 5Department of Oncology, University Hospitals Birmingham, Birmingham B15 2GW, UK; 6Department of Oncology, Royal Surrey NHS Foundation Trust, Guildford GU2 7XX, UK

**Keywords:** kidney cancer, renal cell cancer (RCC), oligometastases, radiation therapy, stereotactic body radiotherapy (SBRT), SABR, stereotactic radiation therapy, primary kidney SABR

## Abstract

**Simple Summary:**

RCC has traditionally been considered radioresistant, with surgery being the gold standard for primary localized RCC. However, not all patients are suitable for surgery and percutaneous, non-surgical options are invasive, with significant limitations. SABR is a non-invasive advanced RT technique that delivers high doses accurately. Growing evidence supports SABR as a definitive alternative therapy for medically inoperable patients, those who decline surgery, are unfit for invasive ablation, or are at high-risk of requiring postoperative dialysis. SABR has shifted the renal radioresistance paradigm, widening the therapeutic window. Additionally, SABR is increasingly used for locally recurrent, oligoprogressive, and oligometastatic disease. This review aims to support the use of SABR across various stages of RCC disease and explore future directions. The overall landscape of RCC is promising, and we are confident that our update will stimulate further research in this field and contribute to the advancement of patient care.

**Abstract:**

Renal cell cancer (RCC) has traditionally been considered radioresistant. Because of this, conventional radiotherapy (RT) has been predominantly relegated to the palliation of symptomatic metastatic disease. The implementation of stereotactic ablative radiotherapy (SABR) has made it possible to deliver higher ablative doses safely, shifting the renal radioresistance paradigm. SABR has increasingly been adopted into the multidisciplinary framework for the treatment of locally recurrent, oligoprogressive, and oligometastatic disease. Furthermore, there is growing evidence of SABR as a non-invasive definitive therapy in patients with primary RCC who are medically inoperable or who decline surgery, unsuited to invasive ablation (surgery or percutaneous techniques), or at high-risk of requiring post-operative dialysis. Encouraging outcomes have even been reported in cases of solitary kidney or pre-existing chronic disease (poor eGFR), with a high likelihood of preserving renal function. A review of clinical evidence supporting the use of ablative radiotherapy (SABR) in primary, recurrent, and metastatic RCC has been conducted. Given the potential immunogenic effect of the high RT doses, we also explore emerging opportunities to combine SABR with systemic treatments. In addition, we explore future directions and ongoing clinical trials in the evolving landscape of this disease.

## 1. Introduction

Radiotherapy (RT) has not traditionally been considered a good treatment option for renal cell cancer (RCC), as it was thought to be intrinsically radioresistant. This view was formed in the 1970s [1,2,3] following studies that showed a lack of benefit from adjuvant and neo-adjuvant RT, delivered at doses that would currently be considered suboptimal. In addition, the kidney was considered to be at risk of radiation-induced long-term damage from relatively low doses of RT [4]. Because of this, RT was relegated to palliative use [5]. 

Until recently, therapy for primary RCC was limited to invasive local interventions, including surgery (open, robotic, or laparoscopic) or percutaneous interventions such as microwave ablation, radiofrequency ablation (RFA), or cryotherapy as the main options.

Surgical resection remains the standard of care (SOC) in localized disease, but this approach is not feasible in all patients due to disease factors, e.g., size, location, and/or patient factors, including co-morbidities and patient choice. Percutaneous techniques also have significant limitations. 

Contrary to the historical notion of radioresistant disease, the current landscape has changed significantly with the advent of stereotactic ablative radiotherapy (SABR). Preclinical models have suggested ablative doses of RT may overcome the hypothetical radioresistance mechanism based on the activation of HIF1α (which is characteristic of clear cell cancers with von Hippel-Lindau [VHL] mutations), which stimulates endothelial cell survival, inducing cell death by alternative means to DNA damage and mitotic catastrophe [5]. 

SABR is a non-invasive advanced RT technique that delivers high doses accurately with a very steep dose gradient over a small number of treatment sessions, usually 5 sessions or less. This approach has recently shown promising local control (LC), an acceptable side-effect profile, and good preservation of renal function (nephron-sparing) in primary RCC for tumors up to 10 cm in diameter, even when these are encroaching on the central portion of the kidney or on the collecting system [6]. It has also been shown to be a safe approach in solitary kidney cases [7]. Given that, its use has become increasingly integrated into multidisciplinary management at different stages of the disease.

A review of clinical evidence supporting the use of ablative radiotherapy in primary, recurrent, and metastatic RCC has been conducted. Given the potential immunogenic effect of the high RT doses, we also explore emerging opportunities to combine systemic targeted agents or immunotherapies with ablative radiation (SABR) as a promising approach that might improve outcomes. In addition, we explore future directions and ongoing clinical trials in the evolving landscape of this disease.

## 2. Non-Surgical Approaches of Primary Renal Cell Cancer (RCC)

Surgery, either partial or total nephrectomy, remains the gold standard local intervention, where possible, in primary localized RCC. However, not all patients are suitable for a surgical approach. Non-surgical options include microwave ablation, RFA, or cryotherapy as the main options but they also present significant limitations and are invasive. They can typically only treat smaller tumors (<4 cm) which need to be distant from the collecting system and vascular structures due to the risk of heat sink effects, which can result in stricture and/or fistula [8].

Some studies have shown the feasibility of other ablative techniques, such as high-intensity, focused ultra-sound ablation and non-thermal, irreversible electroporation. However, these techniques are still considered experimental [9].

Given the limitations of surgery and non-surgical techniques, an ablative, non-invasive alternative was needed.

## 3. The Emerging Role of Stereotactic Ablative Radiotherapy (SABR) for Primary RCC

The technological evolution that RT has undergone has permitted more accurate treatments and consequently higher doses per fraction, while sparing critical surrounding organs at risk (OAR) delivered in ultra-hypofractionated treatment schedules. Kidney SABR has emerged as an ablative approach with curative intent in the primary RCC setting, particularly in the context of patients with primary RCC who are medically inoperable, refuse surgery, or are at high risk of surgical complications (larger renal masses > 4 cm or where the tumor is close to the renal pelvis), who have limited curative treatment options. 

There is a growing body of evidence supporting the use of SABR for primary RCC management, including several retrospective studies, systematic reviews, and meta-analyses, as well as the promising results of the first large prospective trial, which has recently been published [7]. 

In contrast, in the SABR meta-analyses by Correa et al. [10], 26 studies were identified (11 prospective trials), including 383 tumors in 372 patients, most of whom were deemed inoperable. Median follow-up, median age, and mean tumor size were 28.0 (5.8–79.2) months, 70.4 (62–83) years, and 4.6 (2.3–9.5) cm, respectively. Dose fractionation varied, but 26 Gy in one fraction and 40 Gy in five fractions were the most common. The random-effect estimated for LC was 97.2%, grade 3–4 toxicity 1.5%, and post-SABR estimated glomerular filtration rate (eGFR) decrease of 7.7 mL/min. Six patients with pre-existing renal dysfunction (2.9%) required dialysis. Reported toxicity includes mild nausea, fatigue, dermatitis, pyelonephritis, gastric/duodenum ulcer. There was no treatment-related mortality. 

In addition, the International Radiosurgery Oncology Consortium for Kidney (IROCK) [11] published a series of 223 patients with non-metastatic kidney cancer (around 50% with tumors > 4 cm diameter) treated with a single fraction of 25 Gy or multi-fractioned SABR to a median dose of 40 Gy. The five-year LC rate was 94.5%. Most patients had a reduction in eGFR (mean decrease of 5.5 mL/min), with a corresponding rise in creatinine after treatment (mean 28 μmol/L), with six patients requiring dialysis.

Even in cases where the patient only has a single kidney, SABR to the RCC is a potential treatment option [12], showing a 2-year LC of 98% and a change in eGFR ranging from a decrease of 5.8 mL/min to an improvement of 10.8 mL/min. No patients required dialysis.

Furthermore, the International Society of Stereotactic Radiosurgery (ISRS) conducted a systematic review [13] of 36 publications (822 eligible patients). The median LC rate was 94.1% (70.0–100), 5-year progression-free survival (PFS) was 80.5% (95% CI 72–92), and 5-year overall survival (OS) was 77.2% (95% CI 65–89). 

Finally, the TransTasman Radiation Oncology Group (TROG) has recently reported encouraging outcomes of TROG 15.03 FASTRACK II [7,14,15]. This prospective phase II trial included 70 patients with biopsy-confirmed RCC. Patients with inoperable, high-risk, single-lesion kidney tumors or those who declined surgery were enrolled. Treated tumors were relatively large, measuring, on average, 4.7 cm. Patients with tumors < 4 cm (cT1a) received a single fraction of radiation (*n* = 23), and those with tumors > 4 cm (≥T1b) received three fractions (*n* = 47). None of the patients experienced local progression of their RCC during the FU (median 43 months), nor did any patients die from cancer. OS was 99% at 1 year after SABR and 82% at 3 years. One patient experienced a distant recurrence of their cancer. Cancer-specific survival (CSS) for longer than 3 years was 100%. Adverse effects (AEs) were relatively modest, with no grade 4 or 5 toxicities observed. Seven patients (10%) experienced grade 3 AEs, most commonly abdominal pain (*n* = 3). Fifty-one patients (73%) had grade 1 or 2 treatment-related AEs, and 11 patients (16%) experienced no AEs.

Renal function was assessed by measuring eGFR, with a key exclusion criterion being a pre-treatment eGFR < 30 mL/min per 1.73 m^2^. The cohort enrolled had significant pre-existing chronic disease (baseline mean eGFR of 61.1 mL/min per 1.73 m^2^). The average glomerular filtration rate declined by 10.8 mL/min at 1 year and 14.6 mL/min at 2 years after treatment, indicating mild-to-moderate kidney stress. Only one patient required dialysis following treatment. Overall, there was a modest drop in kidney function, which plateaued after 2 years. In fact, in the context of worse baseline renal function and larger tumors compared with surgical trials, the observed mean decline in eGFR in FASTRACK II [7] is similar to the expected decline with partial nephrectomy. The high efficacy rate and the ability to preserve kidney function was attributed to rigorous quality control, as well as the effectiveness of stereotactic radiation.

In line, IROCK recently demonstrated [16] a moderate decline in renal function with a low rate of dialysis at 5 years following SABR. Patients with a solitary kidney (*n* = 56) were compared with those with bilateral kidneys (*n* = 134). The eGFR decreased by −14.5 (SD 7.6) and −13.3 (SD 15.9) mL/min (*p* = 0.67), respectively, with similar rates of post-SABR dialysis (3.6% vs. 3.7%). A multivariable analysis demonstrated that increasing tumor size (OR per 1 cm: 1.57) and baseline eGFR (OR per 10 mL/min: 1.30) were associated with an eGFR decline of ≥ 15 mL/min at one year.

Table 1 is a summary of the most recent studies of SABR in primary localized RCC.

## 4. The Optimal Dose Schedule for SABR in Primary RCC

Currently, there is no single standard dose and fractionation. There are a range of options, which are still a matter of some debate and research (Table 1). However, there is a consensus on administering a biologically equivalent dose in 2 Gy per fraction (EQD2) equal or higher than 72 Gy [13], highlighting that this suggests a dose–effect relationship. Renal function reduces linearly with dose received by the healthy ipsilateral kidney up to 100 Gy BED3. In FASTRACK II [7], the local renal function loss was higher in the 3-fraction cohort than in the single-fraction at 12 and 24 months [23]. Given this, reducing the ipsilateral kidney volume receiving 50% of the dose at the planning stage is recommended to minimize loss of renal function post-SABR treatment. In addition, improving the plan quality overall by using advanced technologies such as better motion management is also recommended [18,22].

The ISRS [13] recommends that optimal dose regimens for SABR in patients with primary RCC include 26 Gy in one fraction if the tumor is ≤4–5 cm and 42–48 Gy in three fractions if the tumor is >4–5 cm, or even 40 Gy in five fractions if the dose constraints for OARs cannot be met for three fractions.

This is in line with the FASTRACK II trial [15], where differences in oncological outcomes between single-fraction and three-fraction SABR schedules were not observed; however, the fractionation schedules represented two different populations based on tumor size.

The Royal College of Radiologists in the United Kingdom have recently published a list of recommended dose/fractionations for a wide range of tumor types. The doses recommended are in line with the above, including 26 Gy in 1 fraction or multiple fraction regimens, such as 42 Gy in 3 fractions/40 Gy in 5 fractions [24].

Table 2 summarizes current recommendations on dosing schedules, and Figure 1 shows some representative cases of SABR for primary RCC.

## 5. Response Evaluation/Follow-Up

To date, LC after SABR is measured using the Response Evaluation Criteria in Solid Tumors (RECIST). This method is still considered the most robust tool for assessing progression following RT but has some limitations with SABR.

In contrast to percutaneous ablation, SABR does not immediately obliterate the tumor and vascular architecture but instead relies primarily upon DNA damage causing loss of proliferative capacity, with ongoing development of cell kill over a prolonged period of time [10]. The typical post-ablation changes, such as immediate histological changes, loss of contrast enhancement, or a complete response, are therefore not seen after SABR. In fact, persistent enhancement following RT is common and not correlated with a risk of subsequent progression in RCC [7]. Furthermore, RCC is a slow-growing tumor, and it can be expected to show a slow radiographic response to treatment.

“Pseudoprogression” is a commonly observed phenomenon in RCC, as the tumor may show an initial size increase that may be misinterpreted as disease progression but may actually be due to treatment-induced inflammation and edema. This eventually stabilizes or results in shrinkage over a prolonged period [25]. To avoid being misled by pseudoprogression, it is best to delay initial imaging after SABR until at least 3 months from completion of treatment [7].

SABR tends to cause a slow decrease in size over months to years, the rate of which does not appear to depend on the dose delivered to the tumor. In the evaluation of response after SABR, the definition of LC is that of stable disease. Ongoing enhancement, or even occasionally increased enhancement, in some cases, can be misinterpreted as persistent or recurrent disease, although no correlation with local failure has been observed [26]. Response assessment after SABR is challenging and demands close monitoring with frequent and long-term imaging follow-up.

Future research is focusing on multiparametric magnetic resonance imaging (MRI) and positron emission tomography (PET), which hold the promise of a better response assessment after SABR [27]. Novel modalities, including functional MRI sequences, showed promise in detecting early response to therapy. In fact, early changes in diffusion and perfusion following renal SABR appear to be a potential predictive imaging biomarker that correlates with subsequent CT–based tumor response.

Biopsy after SABR is not recommended in routine clinical practice and should be considered experimental, as the microscopic evaluation of viable cells can be misleading after exposure to radiation [28].

To conclude, no single follow-up plan is appropriate for all patients; therefore, it should be individualized based on the patient’s requirements [29]. Nevertheless, the general follow-up consensus includes regular cross-sectional imaging, which could be either CT or MRI scans, undertaken initially 3 months after completion of treatment, and then again at 6 and 12 months. After that, scans can be arranged every 6–12 months, as clinically indicated [13,29]. This is in line with what Siva et al proposed, as summarized in the Figure 2.

## 6. The Role of Stereotactic Ablative Radiotherapy (SABR) for Metastatic RCC (mRCC)

Approximately 20% of renal carcinoma patients present with metastatic disease (synchronous) and another 25–40% develop metastasis after radical treatment of the primary (metachronous) [24].

The SOC for metastatic RCC (mRCC) is systemic therapy (ST). Recent breakthroughs in ST, specifically, with immunotherapy (IO) and tyrosine kinase inhibitors (TKIs), have expanded lines of treatment and improved survival. Unfortunately, a cure for mRCC remains elusive, and complete responses are uncommon. Thus, it is important to develop new ST or new combination regimens, maximize the benefit of available therapies, and carefully delay initiation of ST whenever possible.

Metastatic-directed therapy (MDT) historically involved surgery [30] but has recently expanded to include SABR. SABR has emerged as an attractive therapeutic option in the setting of limited metastatic disease (between three and five metastases). Its effectiveness has been demonstrated in prospective clinical trials for selected patients with oligometastases and oligoprogression of different primary histology and lesion locations, both instead of and in combination with ST [31,32,33]. By leaving the tumor in situ, SABR may also induce immune remodeling, both intratumorally and within the systemic metastases, which may augment response to ST [34].

However, in the mRCC scenario, there are still limitations in the published literature, which are mainly retrospective in nature [35] and a lack of high-level evidence from randomized clinical trials. Therefore, the European Society of Therapeutic Radiation Oncology (ESTRO) has recently developed consensus recommendations to provide guidance on the application of SABR in oligometastatic RCC and to highlight the key areas of ongoing debate [36]. Panelists agreed not to apply age or primary RCC histology restrictions for SABR candidates. They established an upper threshold of three lesions to offer ablative treatment in the oligoprogressive setting and agreed on the concomitant administration of immune checkpoint inhibitors (ICI) (within 30 days from SABR). In addition, SABR was agreed upon as the modality of choice for RCC bone and adrenal oligometastases. Although no consensus was reached, there was a major agreement that SABR should be administered on an every-other-day scheme, and it could also represent a useful tool to postpone ST change in selected cases.

## 7. Overview of the Combination of Systemic Therapy (Chemo/IO/Targeted) with SABR for Metastatic RCC (mRCC)

The potential immunomodulatory effect of SABR has led to increased interest in its use in combination with immunotherapy (IO) via the abscopal effect and targeted agents. Preclinical data strongly support this synergy [37], as do emerging results from clinical trials such as RADVAX [38], which reported a 56% response rate of non-irradiated lesions in patients with high-volume mRCC.

In the neoadjuvant setting (neo-SABR), 3 patients with mRCC at diagnosis within a clinical trial (NCT02473536) also showed response in the unirradiated lesions without the concurrent use of ST; therefore, this response was attributable to the abscopal effect. In addition, neo-SABR resulted in decreased Ki-67 and increased PD-L1 expression [39].

In the oligometastatic setting, SABR and a short course of pembrolizumab were evaluated in the RAPPORT trial, which showed an acceptable toxicity profile (13% of G3 and 0% G4–5), with excellent LC (92% at 2 years) [40].

Furthermore, patients receiving ST often develop resistance because of intratumoral mutational heterogeneity and consequently, disease progression. Oligoprogression (OP) means that the progression occurs in a select number of metastases while other lesions remain responsive or stable to a given ST. For these patients, switching to another ST is the SOC. Unfortunately, subsequent lines of therapy are often less effective, and patients ultimately run out of options. In this scenario, administering high doses of RT (SABR) to the drug-resistant sites of progression may overcome this adaptive resistance [39], improving LC while extending the duration of the systemic agent administered. Cheung et al. [41] reported the first prospective trial testing SABR in 37 patients with mRCC receiving tyrosine kinase inhibitors (TKIs) with ≤5 progressing lesions and showed 93% LC and a time-to-change in ST of 12.6 months. Similarly, Hannan et al. [42] treated 37 lesions in 20 patients with oligoprogressive mRCC receiving TKIs, immune checkpoint inhibitors (ICIs), or mammalian target of rapamycin inhibitors (mTORs), with 100% LC, and a time-to-change in ST of 11.1 months.

In contrast, SABR as a monotherapy can be performed as a strategy for postponing the initiation of ST in a low-burden metastatic setting. One such study by Hannan et al. [43] reported a 91% rate of freedom from ST and 100% of LC in mRCC patients with ≤3 mets. Similarly, Tang et al. [44] showed excellent LC and OS rates (97% and 100%, respectively). However, the 1-year PFS was 64%, suggesting that most progressions occur outside the RT field.

With regard to anti-angiogenic TKIs, caution should be exercised when combining these with RT, particularly when the field encompasses a critical tissue such as brain, spinal cord, liver, or small bowel. Although there are no formal guidelines yet, drug interruption would be advisable when delivering higher palliative doses over critical structures [45].

A summary of important clinical trials’ results is listed in Table 3.

Despite the increasing use in routine clinical practice, there is still little information on the safety of combining SABR with modern targeted therapy or IO, and a paucity of high-level evidence to guide clinical management. It is for this reason that consensus such as the OligoCare consortium [47] and further clinical trial results are much needed to guide our daily practice and allow for adequate patient selection.

## 8. SABR as Salvage Treatment for Locoregional Recurrent RCC

Locoregional recurrence after nephrectomy for localized RCC is rare (2–6%) due to improving surgical techniques and adjuvant therapies. However, 5-year survival decreases to 18–46% after recurrence, and the median CSS was 0.7–1 year if left untreated [48].

In this setting, systemic therapy remains the cornerstone approach, and although RT has been investigated for decades, the vast majority of studies have focused exclusively on the intraoperative RT (IORT) [49,50,51,52,53,54].

The selection criteria and efficacy of different treatments are unclear but local therapy could delay disease progression compared with ST alone. In this context, SABR might be effective, but there is a lack of data, which therefore means this is only a theoretical benefit that can be extrapolated from the outcomes of primary tumor SABR.

Only one [48] retrospective series is available, by Yang Liu et al. [48]. In this series, 106 patients with recurrent RCC without distant metastasis were analyzed, 31% of which received ST alone and 69% local therapy (34/73 surgery and SABR −30–40 Gy/5 fractions -in 39/73 patients). Patients receiving local therapy had significantly longer PFS than ST (19.7 vs. 7.5 months, *p* = 0.001). After matching, the PFS in the local therapy group remained higher (23.9 vs. 7.5 months, *p* = 0.001). The median PFS of patients undergoing surgery, SBRT, and ST alone was 15.4, 21.9, and 7.5 months, respectively (*p* = 0.003). The 2-year OS of the local therapy group and ST group was 91.6% and 71.8%, respectively (*p* = 0.084). Local therapy was associated with better PFS (HR 0.37; *p* = 0.0003) and OS (HR 0.23; *p* = 0.002) in multivariate analysis. Grade 2 or higher toxicities related to local therapy occurred in 9/73 patients. Patients undergoing locally directed therapy demonstrated a significantly decreased risk of progression by over 60% (HR 0.37; 95% CI 0.21–0.64; *p* = 0.0003) and risk of death by around 70% (HR 0.23; 95% CI 0.09–0.58; *p* = 0.002).

Therefore, SABR could play a role in the salvage of local recurrence cases with a low toxicity profile and good local control.

## 9. Review of Current Guidelines

Some of the main international clinical guidelines have already included SABR in primary RCC as an alternative therapy in medically inoperable patients [13,29,55]. Even though early results of SABR look encouraging, the latest 2024 European Association of Urology (EAU) update does not yet fully endorse this approach, suggesting that well-conducted prospective studies with longer follow-up are needed [9].

Table 4 shows the recommendations of current guidelines.

## 10. Future Directions/Ongoing Trials

SABR has become the standard treatment for primary RCC in patients unsuitable for other ablative techniques given the potentially high risk of toxicity (>4 cm or proximity to the renal pelvis). The promising outcomes of FASTRACK II [7,14] have shown the potential of this technique with curative intent in a primary setting to be not only an alternative to other local techniques but also a standard alternative to surgery if future studies corroborate its efficacy and toxicity profile. We look forward to seeing the design of randomized trials of SABR versus surgery for primary RCC.

Stereotactic magnetic resonance (MR)-guided adaptive radiotherapy (SMART) may result in more precise treatment delivery through its capabilities for improved image quality, daily adaptive planning, and respiratory motion during treatment with real-time tracking. Thus, a reduction of target volume margins can be applied, and therefore, a reduction in the impairment of kidney function could be achieved. Preliminary outcomes are encouraging [18,22], further results are awaited (MRI-MARK Trial-NCT04580836), and even potential predictive biomarkers of radiosensitivity are being investigated (the ISRAR Database).

In line with the rationale of cytoreductive surgery (CARMENA [57] and SURTIME [58] randomized trials), the cytoreductive role of SABR in mRCC is being investigated, with the advantage not just being a less invasive approach, but also of maintaining the pro-immunogenic and abscopal effect [39,59,60].

The role for SABR in deferring, enhancing, or delaying the switching of systemic therapy upon oligoprogression is another investigational field [60].

It is unclear which patients would derive the most benefit from a particular treatment approach. Thus, identifying, developing, and integrating multimodal biomarkers and driver mutations that predict therapeutic response and resistance is of critical importance to stratify and select suitable candidates. Emerging techniques such as spatial transcriptomics, single-cell analysis, and circulating tumor DNA show tremendous promise in both enhancing our understanding of RCC and uncovering new biomarkers. Advancements in this scenario could further transform the clinical management of RCC, moving toward more personalized and precise therapeutic interventions.

From the above, we can conclude that ongoing trials will help clarify the optimal patient selection, the impact of disease presentation, molecular subclassifications, the optimal combination of therapies, and SABR dosing, fractionations, and sequencing. In the meantime, there is work to be done to implement RCC SABR in our multidisciplinary decision-making framework across different stages of this disease.

Table 5 summarizes the most relevant ongoing trials in kidney SABR.

## 11. Conclusions

The management of RCC is evolving rapidly. SABR represents an attractive and safe option, and this has resulted in its implementation in the therapeutic algorithm in different stages of this disease, alone or in combination with systemic therapy. The promising outcomes of phase II prospective trials are reshaping the landscape of primary localized RCC. The design of randomized trials of SABR versus surgery in this scenario are awaited, but, to date, the available data encourage its implementation as a non-invasive ablative alternative for localized RCC. With metastatic RCC, SABR use has increased in routine clinical practice but there are still unsolved questions about the best candidates, treatment combinations, and schedules. Nevertheless, the overall landscape of RCC is promising, with several ongoing trials, the results of which will help to guide and optimize the diagnosis and treatment tools on a case-by-case basis.

## Figures and Tables

**Figure 1 cancers-16-02678-f001:**
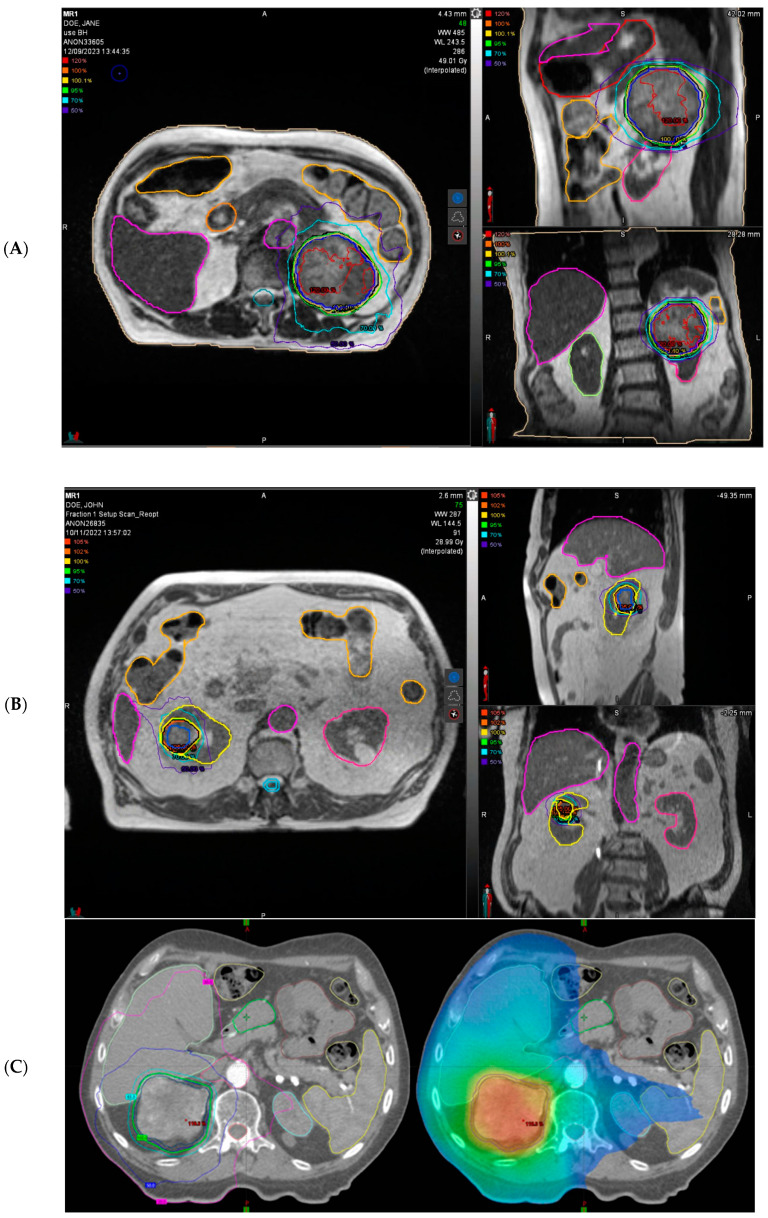
Representative patients’ treatment plan of stereotactic ablative radiotherapy (SABR) for primary renal cell carcinoma (RCC). (**A**) A representative radiotherapy plan treating a large (8.5 cm) left-sided RCC with 40 Gy in five fractions is shown in 3 planes on MR-Linac (ViewRay^®^, Oakwood Village, OH, USA). Planning System MIM^®^ 7.2.8 (Cleveland, OH, USA). (**B**) A representative radiotherapy plan treating a right-sided RCC with 26 Gy in one single fraction is shown in 3 planes on MR-Linac (ViewRay^®^). Planning System MIM^®^ 7.2.8. (**C**) A representative radiotherapy plan treating a right-sided RCC with 42 Gy in 3 fraction (every other day) is shown in 3 planes on conventional linac (Varian^®^ Clinac 2300 iX, Varian Medical Systems, Palo Alto, CA, USA).

**Figure 2 cancers-16-02678-f002:**
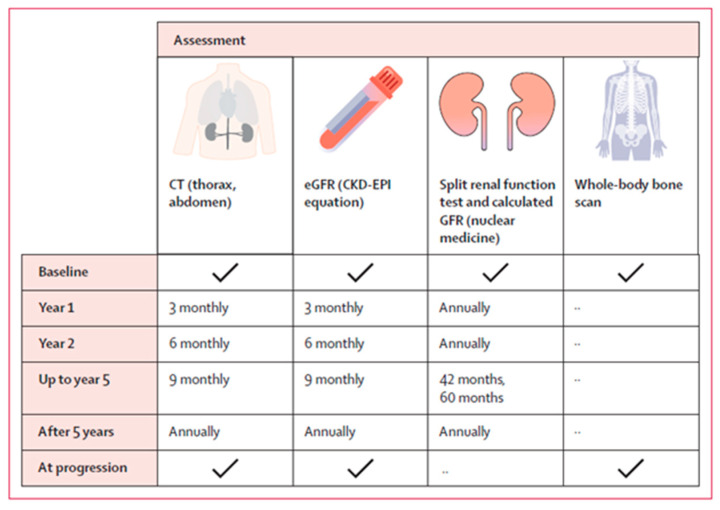
The schedule of assessment proposed by Siva S et al., 2024 [7].

**Table 1 cancers-16-02678-t001:** SABR Studies in primary localized RCC.

Author (Year)	Design	N	Tumor Size (cm)	LINAC	SABR Schedule	Median FU (mo)	Toxicity	Outcomes
Peddada et al. (2020) [17]	Prospective	21	8.7 (4.8–13.8)	CK, fiducials	48 Gy/3 fx (1 pt, 42 Gy/3 fx—mass in renal pelvis)	78	NR	5-y LC 100% OS not reached (78 m FU)
Tetar et al. (2020) [18]	Retrospective	36	5.6 86.1% >4 cm	MRI-Linac	40 Gy/5 fx in 2 weeks	16.4	G2 nausea 1 pt G3 or + 0%	1 y LC 95.7% Mean-eGFR decreased from 55/3 mL/min to 49.3 mL/min after 16 mo. no pt. requiring dialysis
Staehler et al. (2022) [19]	Retrospective	99	2.8	CK	25 Gy/1 fx	28.1	NR	2 y-OS 64% LC 98% −6.5% change eGFR
Siva et al. (2022) [11]	Retrospective	190	4	Various	25–42 Gy in 1–5 fx	60	G1–2: 70% G3: 0% G4–5: 1%	3 y-PFS 72.1% 5 y-PFS 63.6% 5 y-LC 94.5% OS NR −14.2% change eGFR
Lapierre et al. (2023) [20]	Prospective Phase I	13	3.3 (2.3–4)	Standard Linacs (VMAT)	32, 40, or 48 Gy in 4 fx 40 Gy in 5 fx	23	G1–2: 41.7% G3–5 0%	LC 100% −5.9% change eGFR
Hannan et al. (2023) [21]	Prospective Phase II	16	3.2 (2.6–3.9)	Varian Truebeam or Vitalbeam, ElektaVersa	36 Gy in 3 fx 40 Gy in 5 fx	36	50% G1–2	3-y OS 79% LC 94% −12.1% change eGFR
Yim et al. (2023) [22]	Prospective Phase I	20	4.4 (3.5–5.9)	MRI-Linac	40 Gy in 5 fx	17	G3 + 0%	A single patient developed local failure at 30 mo.
Siva et al. (2023) [7,14,15] FASTRACK II	Single arm Prospective Phase II	70	4.6 (3.7–6.0)	Standard Linacs (VMAT)	≤4 cm (cT1a) 26 Gy/1 fx >4 cm, 42 Gy/3 fx (48 h apart)	43	73% G1–2 10% G3 0% G4–5	100% LC (m-FU:43 m) 1 y-OS 99% 3 y-OS 82% 1 pt. distant recurrence

N: number of patients; LINAC: linear accelerator; FU: follow-up; mo: month/s; fx: fraction/s; LC: local control; PFS: progression-free survival; OS: overall survival; NR: not reported; eGFR = estimated glomerular filtration rate; CK: Cyberknife^®^; VMAT: volumetric modulated arc therapy.

**Table 2 cancers-16-02678-t002:** Commonly used dose and prescription.

Total Dose	Fractions	Dose per Fraction	EQD2 (α/β 10)	Tumor Size	Duration
26 Gy	1	26 Gy	93.6 Gy	≤4–5 cm	Single fraction
42 Gy	3	14 Gy	100.8 Gy	>4–5 cm	Non-consecutive days (48 h apart).
48 Gy	3	16 Gy	124.8 Gy	>4–5 cm	Non-consecutive days (48 h apart).
40 Gy	5	8 Gy	72 Gy	if OAR constraints cannot be met for 3 fx.	Non-consecutive days (48 h apart).

**Table 3 cancers-16-02678-t003:** Studies of SABR in patients with metastatic renal cancer (mRCC).

Author (Year)	Design	N	Patient Group	SABR Schedule	Drug	FU (m)	Toxicity	Outcomes
SABR in monotherapy								
Tang et al. (2021) [44]	Prospective single arm	30	omRCC ≤ 5 mets, at most 1 prior ST	SABR to defer ST 60–70 Gy/10 fx 52.5–67.5 Gy/15 fx	No ST	18	Grade 3: 6.6% Grade 4: 3.3%	Median PFS: 23 mo, 1-year PFS 64% 97% local control. 1-year OS 100%
Hannan et al. (2022) [43]	Prospective single arm	23	omRCC ≤ 3 mets, no prior ST	SABR to defer ST	No ST	21.7	No Grade 3–4, 1 death from colitis possibly SABR-related	Freedom from ST at 1 year 91%. 100% Local control
SABR in combination with targeted therapy and immunotherapy								
Buti et al. (2020) [46]	Retrospective	48 (57 mets)	omRCC ≤ 5 mets, or OP RCC (≤3 mets) (brain lesions excluded)	5 Gy x 5 fx 6 Gy x 5 fx 8 Gy x 3–5 fx 10 Gy x 3–5 fx 15 Gy x 3 fx	concurrent ST: 28 pt Interrupted ST 29 pt Reintroduction ST: 30 pt	32	G1–2: 6% G3–4: 0%	1-year LC 83.6% 2-year LC 72.4% w/ST: PFS 28.9 m OS 49.2 m 1-year OS: 93.7% 2-year OS: 84.9%
Hammers et al. (2020) RADVAX [38]	Prospective single arm Phase II	25	High metastatic volume. Prior ST with TKI and IL2 were allowed.	50 Gy/5 fx to 1–2 disease sites. Delivery between the 1st and 2nd dose of N/I	Dual ICI (N/I)		Grade 2: 8% (SABR-field pneumonitis)	ORR of non-irradiated lesions 56% (by RECIST 1.1)
Cheung et al. (2021) [41]	Prospective single arm	37	mRCC on TKI w/≤5 OP mets	SABR to maintain ST in pt w/oligoprogressive mRCC	TKI	11.8	No ≥ grade 3	93% Local Control, Time to change in ST 12.6 months
Siva et al. (2022) RAPPORT [40]	Prospective single arm Phase I/II	30 (83 mets)	omRCC ≤ 5 mets, SABR+ 8 C of Pembro (≤2 lines of prior ST)	SABR + ICI 20 Gy/1# (77%) 30 Gy/3# (23%)	ICI (Pembrolizumab, 6 months)	28	Grade 3: 13% Grade 4–5: 0%	2-year LC: 92% 1-year PFS 60%, 2-year PFS 45%, 1-year OS 90% 2-year OS 74%
Hannan et al. (2022) [42]	Prospective single arm	20 (37 nets)	mRCC w/≤3 OP mets	SABR to maintain ST 25 Gy/1 fx (16.2%) 36 Gy/3 fx (56.8%) 40 Gy/5 fx	TKI (40%) ICI (40%) ICI + TKI (15%) mTOR-I + TKI (5%)	10.4	Grade 3: 5% Grade 4–5: 0%	100% Local Control Time to change in ST 11.1 months 1-year OS 73.7%

mRCC, metastatic renal cell carcinoma; omRCC, oligometastatic renal cell carcinoma; OP: oligoprogressive; OS, overall survival; PFS, progression-free survival; RCC, renal cell carcinoma; SABR, stereotactic ablative radiotherapy; m: months; FU: follow-up; ST: systemic therapy; C: cycle; TKI, tyrosine kinase inhibitor; ICI: immune checkpoint inhibitor; N/I: nivolumab + ipilimumab; mTOR: mammalian target of rapamycin inhibitor; ORR: objective response rate.

**Table 4 cancers-16-02678-t004:** Review of current guidelines.

Guidelines	Recommendations		Category/Strength of Recommendation	Level of Evidence
EAU [56]	Offer SABR for clinically relevant bone or brain metastases for local control and symptom relief.		Weak	3
ESMO [55]	Unresectable local or recurrent disease with the aim of improving LC. For patients in whom surgery cannot be carried out due to poor PS or unsuitable clinical condition or if other local therapies such as RFA are not appropriate.		B	IV
	Oligometastatic, oligoprogression, or in mixed response scenarios with immunotherapy or targeted therapies.		B	IV
	Modern image-guided techniques are needed to enable a high biological dose.		B	IV
NCCN^®^ [29]	Primary treatment for medically inoperable patients (not optimal surgical candidates) with kidney cancer	stage I	2B	
		stage II/III	3	
	Oligometastatic disease	Stage IV	2A	
ISRS [13]	Optimal dose regimens for SABR in patients with primary RCC include 26 Gy in one fraction if the tumor is ≤4–5 cm and 42–48 Gy in three fractions if the tumor is >4–5 cm, or potentially 40 Gy in five fractions if the dose constraints for OAR cannot be met for three fractions.		Moderate	IV
	A routine post-SABR biopsy should not be performed to evaluate response and is only recommended in patients with imaging findings concerning for disease progression.		Strong	IIb
	For patients with a solitary kidney, SABR is an approach associated with both excellent local control and acceptable renal function preservation (except in patients with stages 4 and 5 chronic kidney disease); technical approaches to reduce the volume of irradiated kidney, particularly in the intermediate dose-wash region, is recommended.		Strong	IIIa
	Optimal post-treatment follow-up schedule after SABR for primary renal cell carcinoma includes cross-axial imaging of the abdomen, including both kidneys and adrenals, every 6 months and surveillance scans, including chest imaging at a minimum.		Moderate	IIb

EAU: European Association of Urology; ESMO: The European Society for Medical Oncology; NCCN^®^: The National Comprehensive Cancer Network^®^; ISRS: The International Society of Stereotactic Radiosurgery; SABR: stereotactic ablative radiotherapy; LC: local control; RCC: renal cell carcinoma; OAR: organs at risk; PS: performance status; RFA: radiofrequency ablation.

**Table 5 cancers-16-02678-t005:** Main ongoing studies.

Trial	N	Design	Primary Endpoint	Status/Estimated Completion Date
Localized primary RCC				
NAPSTER (NCT05024318)	26	Randomized to neoadjuvant SABR 42 Gy/3 fractions followed by nephrectomy, versus neoadjuvant SABR + pembrolizumab followed by nephrectomy in patients with localized primary RCC	Evaluating changes in tumor-responsive T-cells following neoadjuvant-mPR* post-SABR with or without pembrolizumab * mPR rate is defined as <10% residual tumor at post-nephrectomy -CD8+ -TIL	Recruiting/ December/2024
MRI-MARK Trial (NCT04580836)		Phase II. MRI-guided SABR in patients with early-stage kidney cancer	1st: 2-year LC 2nd: preservation of renal function, G3 AEs, MRI biomarkers, pCR by biopsy, OS and DRFS.	Unknown
ISRAR Database (NCT06041555)	600	Prospective Observational. MRgRT and Radiobiological Data (Irm Sequences for Radiobiological Adaptative Radiotherapy) * Allowed prostate, kidney, cervix, head and neck cancer, or glioblastoma conditions.	Hypoxia mapping: Generate 3D maps of intra-tumoral hypoxia and characterize their evolution during treatment, thanks to the establishment of a prospective database of MRI sequences	Recruiting/2029-01-08
Metastatic				
SABR to the primary				
SAMURAI (NCT05327686) [61]	240	Randomized Phase II for metastatic RCC with intact primary tumor. SAbR 42 Gy/3 # to the primary and SOC IO (ICI-doublet), vs. ICI-doublet alone. A variety of combination IO and targeted therapies are allowed on the trial.	1st: PSF 2nd: Best overall response, OS, 2nd line therapy-free survival, cytoreductive nephrectomy, Treatment-free survival, G3 + AEs.	Recruiting/2028
CYTOSHRINK [62] (NCT04090710)	78	Phase II randomized. The addition of cytoreductive SABR (30–40 Gy/5) to the primary mass + SOC combination I/N vs. I/N alone for the treatment of metastatic kidney cancer.	1st: PFS 2nd: Safety, OS, Response Rate, QoL and Drug tolerability	Recruiting/ March/2024
SABR to the mets				
ASTROs (NCT06004336)	144	Phase II randomized to definitive SABR followed by maintenance pembrolizumab versus definitive SABR followed by observation for oligometastatic RCC	1st: PFS 2nd: OS, time to next ST line after Pembro. LRFS, DRFS and AEs	Recruiting/ January/2028
EA8211-SOAR [63] (NCT05863351)	472	Phase 3 randomized noninferiority. SAbR to the mets followed by SOC ST vs. SOC ST alone in patients with oligomet (2–5) RCC. * If primary is intact, local treatment must be performed before randomization.	1st: OS & AEs (CTCAE) 2nd: PFS & PFS-start of ST Testing whether SABR can be used to defer systemic therapy	Recruiting/ 2030
Oligoprogressive				
NCT04974671		Single-arm phase II. SABR for Oligoprogression (1–5 lesions) on ICI in mRCC	PFS	Recruiting April/2027
GETUG-StORM-01 (NCT04299646)	114	Randomized 2:1 Phase II. Patients with 1–3 oligoprogressive sites (mRCC) under either TKIs or ICIs, randomizing patients to continuation of ongoing ST +/− SABR on all progressive metastatic lesions	1st: PFS 2nd: AEs, LC & OS.	Recruiting July/2024

Vs: versus; IO: immunotherapy; ST: systemic treatment; SOC: standard of care; MRI: magnetic resonance imaging; SABR: stereotactic ablative body radiotherapy; pCR: pathologic complete response; I/N: ipilimumab and nivolumab; LRFS: local recurrence free survival; DRFS: distant recurrence free survival; AEs: adverse events; OS: overall survival; QoL: quality of life.

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
