# Peer review of "The Landscape of Stereotactic Ablative Radiotherapy (SABR) for Renal Cell Cancer (RCC)"

_cancers, 2024, doi:10.3390/cancers16152678_

Round 1

Reviewer 1 Report

Comments and Suggestions for Authors

The management of RCC is evolving rapidly. SABR represents an attractive and safe option, leading to its incorporation into the therapeutic algorithm at different stages of the disease, both alone and in combination with systemic therapy. The promising outcomes of phase II prospective trials are reshaping the landscape of primary localized RCC. Randomized trials comparing SABR to surgery in this scenario are awaited, but the available data to date encourage its use as a non-invasive ablative alternative for localized RCC. In metastatic RCC, the use of SABR has increased in routine clinical practice, although there are still unresolved questions about the best candidates, treatment combinations, and schedules. Nevertheless, the overall outlook for RCC is promising, with several ongoing trials whose results will help to guide and optimize diagnosis and treatment on a case-by-case basis.

The authors wrote a well-organized, thorough, and insightful review article. Only minor grammatical errors and punctuation were detected.

Comments on the Quality of English Language

  • 1. Improve sentence flow and structure for better readability.
  • 2. Fix minor grammatical errors and punctuation.

Author Response

REVIEWER 1   

  • 1. Improve sentence flow and structure for better readability.
  • 2. Fix minor grammatical errors and punctuation.

Thank you for pointing this out. The manuscript’s flow and structure have been thoroughly reviewed by a native English speaker (co-author). In addition, grammatical errors and punctuation have been amended. 

Reviewer 2 Report

Comments and Suggestions for Authors

It is a interesting review for SABR for renal cell carcinoma  with primary lesion , local recurrence site or metastatic site management. However, it needs some revision and clarification.

1. In the abstract line 32:  unsuitable to ablation ..... , It is better to define , invasive ablation ( surgery or percutaneous ... )

2.  line 39 :" In addition, we explore future directions and ongoing clinical trials in the evolving landscape of this disease."    This review aims to evaluate  SABR  for RCC.  The other treatment should be mentioned only in comparison with SABR.  It does not need to pay much space to deal with the evolving landscape in RCC.

3. Keyword should be according to MeSH .

4. In line 51 : in  addition, the kidney was considered to be at risk of radiation induced long-term damage from relatively low doses of RT.      The rationale in this sentence seems not to be good.  Hoe about high dose RT ?

5. In line 60-65 .     RCC  should still  be classed to  radio-resistant  or radio-sensitive  only to SABR ?   

6. Line 75-80. Nearly copy and paste Line 35-40 

7. Line 110  , Typo  to candidates  

8. Line 125.  98.4 %   ?? Two years LC or 4 years LC ?

9.   Line 132-133 Furthermore, the International Society of Stereotactic Radiosurgery (ISRS) conducted 132 a systematic review "( (15)) " of 3972 publications and 36 studies (822 patients).   I can not understand the sentence meaning. 

10. Table I ; please explain   LINAC in the foot note 

11.  In 5. Response Evaluation/Follow-up:

Please precisely define  how to define treatment failure  ?

 When to do image study after treatment?   one month or 3 months after treatment ?  what kind of image should be chose ?   

12. Line 367 sequentdecrease   Typo ?

ndont 

Author Response

REVIEWER 2: It is an interesting review for SABR for renal cell carcinoma with primary lesion, local recurrence site or metastatic site management. However, it needs some revision and clarification.

Dear reviewer 2,

We really appreciate your inputs in order to improve our manuscript. Please, find enclosed the responses:

  1. In the abstract line 32:  unsuitable to ablation ...., It is better to define, invasive ablation (surgery or percutaneous ...). Thank you. It has been amended.
  2. line 39:" In addition, we explore future directions and ongoing clinical trials in the evolving landscape of this disease."    This review aims to evaluate SABR for RCC.  The other treatment should be mentioned only in comparison with SABR.  It does not need to pay much space to deal with the evolving landscape in RCC.

We agree with the reviewer. Nevertheless, considering that this is an emerging indication in primary RCC, and given the unresolved questions regarding the metastatic scenario, we found it worthwhile to include subheading 10. This section highlights ongoing trials that will hopefully clarify the optimal management of these patients and we believe it is of interest to the scientific community.

  1. Keyword should be according to MeSH.

Corrected as follows:

-Kidney cancer is registered into MeSH

-Renal cell cancer (RCC) (instead of carcinoma)

-Oligometastases or similar have not been found, and these authors considered it a needed concept.

-Radiotherapy has been changed for Radiation Therapy

-Stereotactic Body Radiotherapy (SBRT) inserted. We strongly recommend that you keep the acronyms that are given as the way to name it in routine practice.

-Stereotactic Radiotherapy has been replaced by Stereotactic Radiation Therapy according to MeSH.

  1. In line 51: in addition, the kidney was considered to be at risk of radiation induced long-term damage from relatively low doses of RT. The rationale in this sentence seems not to be good.  Hoe about high dose RT?

Historically, misconceptions and fears about kidney injury from radiotherapy limited its use to palliative treatments at low doses. As Dawson et al. (4) explained in their manuscript, the pathophysiology of radiation injury was poorly understood, and predictive damage models were suboptimal.

Nowadays, long-term renal function outcomes following SABR for primary RCC are now available. moderate renal function decline and low dialysis rates even in patients with a solitary kidney have been reported. SABR thus represents a promising non-invasive, nephron-sparing option for patients with localized renal cell carcinoma.

In order to support it with more solid data, we have added a recent publication from IROCK (https://doi.org/10.1016/j.euo.2024.06.012)--reference number 18

We hope it is now clearer for you.

  1. In line 60-65.     RCC should still be classed to radio-resistant or radio-sensitive only to SABR?   

The concept lies in the tumour’s histology. This radioresistant concept was formed in the 1970s based on doses that would currently be considered suboptimal.

However, SABR, with its high doses, can overcome this radioresistance and effectively achieve tumour response.

  1. Line 75-80. Nearly copy and paste Line 35-40 

Indeed, but as the abstract summarizes the major aspects of the entire paper, it is logical that some parts are quite similar. We have modified it but the essence of the message has been maintained.

  1. Line 110, Typo to candidates -- Thank you. It has been amended in the manuscript
  2. Line 125.  98.4 %?? Two years LC or 4 years LC? Amended.
  3.  Line 132-133 Furthermore, the International Society of Stereotactic Radiosurgery (ISRS) conducted 132 a systematic review “((15)) " of 3972 publications and 36 studies (822 patients).   I cannot understand the sentence meaning. 

Amended: “36 publications (822 eligible patient)”

  1. Table I; please explain   LINAC in the foot note –Thank you. It has been included in the manuscript
  2. In 5. Response Evaluation/Follow-up:

Please precisely define how to define treatment failure? Local failure is considered evaluated radiographically by RECIST criteria version 1.1 principles (progressive disease -increase in tumour size by 20%) at time >6 months (to exclude pseudo-progression phenomenon) after completion of treatment.

When to do image study after treatment?   one month or 3 months after treatment?  what kind of image should be choosing?  CT scan is appropriate if the renal function allows. If not, MRI is also appropriate, ideally the same image modality as was available pre-SABR to allow a more complete comparison. I would recommend first image at 3 months.

  1. Line 367 sequentdecrease   Typo? -- Thank you. It has been corrected in the manuscript

Reviewer 3 Report

Comments and Suggestions for Authors

The topic is interesting even if the role of SBRT on primary disease remains still debatable. 
On the other hand, there is a crucial role in oligometastatic disease to achieve a no evidence of diseade status and avoid systemic treatments or delaying a shift to a subsequent line of therapy. Consider reference PMID 37444442.

Author Response

REVIEWER 3: The topic is interesting even if the role of SBRT on primary disease remains still debatable. On the other hand, there is a crucial role in oligometastatic disease to achieve a no evidence of disease status and avoid systemic treatments or delaying a shift to a subsequent line of therapy. Consider reference PMID 37444442.

Thank you for your input. Your suggested reference has been included (ref number 32)

Round 2

Reviewer 2 Report

Comments and Suggestions for Authors

accept after revision